# Sustainable Coating Paperboard Packaging Material Based on Chitosan, Palmitic Acid, and Activated Carbon: Water Vapor and Fat Barrier Performance

**DOI:** 10.3390/foods11244037

**Published:** 2022-12-14

**Authors:** Jackson Wesley Silva dos Santos, Vitor Augusto dos Santos Garcia, Anna Cecilia Venturini, Rosemary Aparecida de Carvalho, Classius Ferreira da Silva, Cristiana Maria Pedroso Yoshida

**Affiliations:** 1Institute of Ambiental, Chemical and Pharmaceutical Sciences, Federal University of São Paulo, Diadema 09913-030, Brazil; 2Department of Technology, State University of Maringá, Umuarama 87506-370, Brazil; 3Faculty of Animal Science and Food Engineering, University of São Paulo, Pirassununga 13630-000, Brazil

**Keywords:** coating, paperboard, eco-friendly packaging

## Abstract

Synthetic polymer coatings impact the biodegradable behavior of cellulosic packaging material. The environmental consequences of food packaging disposal have increased consumer concern. The present study aimed to use natural polymer coatings incorporating palmitic acid and activated carbon applied to paperboard surfaces as a sustainable alternative to improve cellulosic packaging material’s moisture and fat barrier properties, minimizing the environmental impact. The coating formulation was defined using a Factorial Experimental Design with independent variables: chitosan, palmitic acid, activated carbon concentrations, and the number of coating layers. The highest concentration of chitosan (2.0% *w*/*w*) filled the pores of the cellulosic paperboard network, supporting the compounds incorporated into the filmogenic matrix and improving the fat resistance. The water vapor permeability of the coated paperboard material (range: 101 ± 43 to 221 ± 13 g·d^−1^·m^−2^) was influenced by the hydrophobicity effect of palmitic acid, the non-polar characteristic of activated carbon, and the number of applied layers. The coating formulation selected was a chitosan concentration of 2.0% (*w*/*w*), a palmitic acid concentration of 1.8% (*w*/*w*), an activated carbon concentration of 1.2% (*w*/*w*), and an application of three layers. The coating provides the potential for a paperboard surface application, improving the cellulosic packaging material’s fat and moisture barrier properties and maintaining biodegradability and recyclability.

## 1. Introduction

Synthetic polymers (plastic) are the most widespread material used in the food packaging sector [1] due to their excellent barrier properties, mechanical durability, and versatility [2]. Many of these plastic packaging are used only once and are known as single-use plastics (SUPs) [1]. The continued disposal of packaging material has created an unsustainable worldwide scenario in environmental waste. SUPs lose more than 95% of their economic value after a single use. In many cases, after consumption of the main product, SUPs are incorrectly discarded directly into nature. The impact of SUPs needs urgent alternatives to protect human health and ecological systems [2].

Cellulose-based packaging has been widely used and has gained prominence in its application due to sustainability, biodegradability, and recyclability [3]; however, its hydrophilic nature directly influences the barrier to water vapor and its mechanical properties [4]. Coatings are applied to the paper surface to improve these properties, but the synthetic coatings compromise the biodegradability process of the paper, turning it into a polluting material [5].

Multilayer packages are a junction of different materials, adding the properties of each to improve the material’s performance as a barrier to oxygen, air, light, moisture, aroma, mechanical properties, and others [6,7,8]. Aluminum is used in some systems as a layer to act as a barrier (gas, light, and microorganism) in packaging for snacks and aseptic products due to its metallic crystalline structure and very low porosity [9,10,11]. However, aluminum-based materials reduced the material’s recyclability [12] due to the high cost of separating material layers [9,12]. In addition, recycling processes generally are intended for single-component materials [6].

Sustainable alternatives were studied, such as natural polymer coating (polysaccharides and proteins) [13,14,15]. Chitosan is a natural polymer, a polysaccharide obtained from the deacetylation of chitin, which is found in insect exoskeletons, some fungi cell walls, and crustaceans shells [16]. Chitosan film or coating showed biocompatibility, antimicrobial properties, low toxicity, biodegradability, selectivity, and gas barriers, such as oxygen (O_2_) and carbon dioxide (CO_2_), water and fat barrier properties, and mechanical resistance [17,18]. According to Brodnjak [19], blends of chitosan and curdlan in the ratio of 5:5 as a paper coating improved the material’s mechanical properties when compared to only chitosan or curdlan coatings.

Natural polymers have a hydrophilic nature. Different lipids were incorporated into natural polymer films to decrease the affinity of the coating matrix with water molecules [20,21], such as carnauba [22], bee [23], and candelilla [24] waxes, and fatty acids, such as palmitic [25] and stearic [26] acid. Palmitic acid is a fatty acid found in plants and animals. Palm oil presents around 44% of palmitic acid, and its chemical structure has 16 carbon atoms [27,28]. The presence of palmitic acid in chitosan films and coatings reduced the interaction of the polymeric matrix with water molecules [29].

The humidity and gas permeation, such as oxygen, carbon dioxide, and ethylene, can also be restricted in packaging material by an active coating on the packaging surface [30,31]. Activated carbon and zeolite are adsorbent agents that regulate the gas concentrations inside food-packed products [32]. Activated carbon is obtained by the thermal decomposition of carbonaceous materials, such as coconut straw and pinewood and presents a highly porous structure [33]. Incorporating a gas-controlling agent directly into a polymer matrix that forms the packaging material allows for a larger area for gas control [31]. Active paper packaging coated with chitosan containing zeolite and vanillin regulated the ethylene gas in the ripening of mango, due to ethylene adsorption on the surface of the zeolite, in addition to the antifungal action of vanillin decreasing the anthracnose appearance (a disease that affects the mango during its storage) [34].

This work aimed to develop and characterize a sustainable alternative material based on paperboard packaging coated with chitosan, palmitic acid, and activated carbon applied in several layers, and to evaluate its performance as a water vapor and fat barrier. The coating formulation was defined using a Factorial Experimental Design with independent variables: chitosan, palmitic acid, activated carbon concentrations, and the number of layers. The motivation of this work was the natural and easy biodegradation of the raw materials used in coating formulations that are entirely renewable and to reduce the environmental impact caused by the disposal of packaging materials, improving the performance of the functionalities of paperboard-based packaging.

## 2. Materials and Methods

### 2.1. Materials

Chitosan with a degree of deacetylation of 82% and an average molar mass of 1.47 × 10^5^ g mol^−1^ was from Polymar (Brazil). Powdered activated carbon (31616) was from Sigma Aldrich (Brazil) and palmitic acid was from Synth (Brazil). Sheets of Duplex Cardboard Paper with a grammage of 250 g·m^−2^ were purchased at a local market in São Paulo (Brazil).

### 2.2. Experimental Factorial Design–Coating Formulation

The determination of the material coating formulation followed the Experimental Factorial Design 2^4^ [35]. Four independent variables were evaluated: chitosan concentration (C_CH_), activated carbon concentration (C_AC_), palmitic acid concentration (C_PA_), and the number of layers applied (NL, 1 L—one layer, 3 L—three layers, and 5 L—five layers). The experimental design matrix consisted of 16 combinations (2^4^) and three repetitions of the central point [35]. Table 1 presents the levels studied in each of the independent variable of the Factorial Experimental Design 2^4^ + central points (level 0).

### 2.3. Chitosan, Palmitic Acid, and Activated Carbon Coating Dispersion

The chitosan, palmitic acid, and activated carbon coating dispersion were prepared following Yoshida, Oliveira, and Franco [29]. The chitosan dispersion was prepared at different concentrations in an acidic medium. Acetic acid was added stoichiometrically, considering the chitosan’s mass and degree of acetylation. The suspension was homogenized in a magnetic stirrer for 60 min. According to the Experimental Design, palmitic acid and activated carbon were incorporated at different concentrations under rigorous homogenization at 20,000 rpm (UltraTurrax homogenizer, T25, IKA, Werke, Niehl, Germany) for 10 min.

### 2.4. Paperboard Coating

The coating was prepared as proposed by Maciel, Franco, and Yoshida [36]. The paperboard sheets were coated with the suspension using a stainless-steel extender of 150 µm (Regmed, Osasco, Brazil) and dried in a forced convection oven (Marconi, MA 035/100, Piracicaba, Brazil) at 150 °C for 120 s. The coating process was repeated to obtain each experiment’s desired number of layers.

### 2.5. Coating Homogeneity

Coating homogeneity was evaluated in triplicate using the colorimetric test adapted from Marcy [37]. A 0.5% erythrosine dye solution in isopropanol was applied to the coated part of the paper (15 cm × 20 cm) with cotton and tweezers. The entire surface was covered with the colored solution. The samples were kept upright and dried at 50 °C for 1 min in an oven (Marconi, MA 035/100, Piracicaba, Brazil). The dried material was visually examined opposite the coating.

### 2.6. Characterization of Coated Paperboard

#### 2.6.1. Thickness, Grammage, and Coating Weight

The thickness (μm) of the coated paperboard was measured using a digital manual micrometer (Mitutoyo, MDC-25M, Miyazaki, Japan). Measurements were performed at five different points in each sample, and five specimens were used per test, according to ASTM D645 [38]. The grammage (g·m^−2^) followed the method ASTM D646-96 [39], ten samples were cut in the dimension of 12.5 cm × 12.5 cm, and the mass was measured with an analytical balance (Ohaus, ARC 120, Barueri, Brazil). The weight of the coatings (CW, g·m^−2^) was theoretically estimated based on the applied coating mass and the application area.

#### 2.6.2. Water Absorption Capacity (Abs–Cobb Test)

The water absorption capacity (Abs) was determined according to standard D3285-93 [40]. Ten samples with 12.5 cm × 12.5 cm were cut and placed in a desiccator for 72 h at room temperature (25 ± 2 °C) with a controlled relative humidity (50% RH). Each sample was weighed on an analytical balance (Ohaus, ARC 120, Barueri, Brazil), fixed in the Cobb equipment (Regmed, Osasco, Brazil), and 100 mL of distilled water was added to come into contact with the surface delimited by the device ring for 120 s. The sample was removed, and the excess water was removed with absorbent paper. The sample was immediately weighed. The results are expressed in g·m^−2^.

#### 2.6.3. Water Vapor Transmission Rate (WVTR)

WVTR was determined using the standardized methodology ASTM E96/E96M-10 [41]. Five material samples were cut into a disk shape over a permeation capsule, fixed in cells containing silica, and then placed in desiccators at a controlled relative humidity (RH) equal to 50% for 72 h. The results are expressed in g·d^−1^·m^−2^.

#### 2.6.4. Fat Resistance

The test followed the methodology TAPPI pm-96 (T559) [42]. Test solutions (Kit numbers 1–12) were prepared with different concentrations of ricin oil, toluene, and n-heptane. A drop of the solution was applied to the material’s surface for 15 s, the excess solution was removed, and the appearance or not of a stain on the back of the paperboard sheet was verified. The solution with the highest Kit number, which remained on the surface of the package without causing stains on the back, was adopted as the fat-repellency value.

#### 2.6.5. Microstructural Analysis

The methodology followed Reis et al. [43]. Samples were covered with gold and were evaluated in a scanning electron microscope LEO 440i (LEO Electron Microscopy, Oxford, Cambridge, England) at 50 kV, with 1000× magnification on the surface and cross-section.

### 2.7. Statistical Analysis

The results of the Factorial Experimental Design were analyzed using Statistica 13.5 and statistically evaluated using the analysis of variance (ANOVA) and mean of the effects.

## 3. Results and Discussions

### Characterization of Coated Paperboard

The dispersion of chitosan, palmitic acid, and activated carbon was formed. The chitosan, palmitic acid, activated carbon concentration ranges, and the number of layers to be applied were determined according to several preliminary tests carried out throughout the beginning of the study. The coated paperboard visually showed homogeneity, good adhesion, and no delamination even after continuous handling. The coatings with high palmitic acid concentration (1.8% *w*/*w*, +1), an increased number of layers (5 L, +1), and a low concentration of chitosan (1.0% *w*/*w*, −1) presented a non-homogeneous visual aspect.

Table 2 presents the coating formulations developed according to the Factorial Experimental Design 2^4^ + central points, the nomenclature adopted for the materials in each test, and the estimated coating weight (CW) applied to the paperboard surface. Table 2 also presents the results for thickness, grammage, water absorption capacity (Abs), water vapor transmission rate (WVTR), and fat resistance (Kit number) for each material produced.

The coating homogeneity was evaluated in each material developed according to the formulations defined by the Factorial Experimental Design 2^4^ + central points. Figure 1 shows the back of the paperboard sheet surfaces where it is possible to verify the presence of any spots of the colorimetric solution that has permeated through the coated paperboard matrix. In all the coating formulations, the coating homogeneity was observed due to a lack of colored spots present on the back of the paperboard surface. The colorimetric solution did not permeate the cellulosic matrix due to the presence of the coating. The homogeneous coatings were also observed by Gatto et al. [44] in chitosan with different degrees of acetylation coatings applied in three layers on the paperboard surface. The high solid content of the coating filled the cellulosic matrix’s pores, improving the paper’s surface [43].

Table 3 presents the average of the effects of the independent variables studied (C_CH_, C_AC_, C_PA,_ and NL), indicating whether the variation from the −1 to the +1 level of the independent variables caused a significant positive or negative change in response to the evaluated properties: thickness, grammage, water absorption capacity, and water vapor transmission rate. The C_CH_, C_AC_, C_PA_, and NL effects were important in defining the coating formulation to obtain efficient properties for food packaging. Increasing the C_PA_ from 0.2% (−1) to 1.8% (+1) and the NL from 1 L to 5 L, caused the thickness to increase significantly to 42 and 35 μm, respectively. The thickness increment was associated with the biopolymeric filmogenic penetration through the cellulosic paper matrix [45,46]. The C_CH_, C_PA_, and NL positively and significantly affected the grammage. The NL promoted the most significant effect varying from 1 L (−1) to 5 L (+1). The increase in grammage could indicate the effectiveness of the coating film’s formation on the material’s surface [47]. The variation in the number of layers applied, from 1 L to 5 L, may have influenced the statistical response of grammage, generating a lack of fit and low R^2^ value, as can be seen in Table 3.

The water absorption capacity (Abs) is related to the resistance of the cellulosic material in contact with water [43]. There is no universal standard for comparison of the Cobb test, as it depends on the specific application of the material [48]. Abs generated a significative and predictive model, described by Equation (1).
(1)$$\mathrm{Abs}\;\left(g\cdot m^{-2}\right)=45+7.5625C_{CH}-2.5625C_{PA}+0.0625C_{AC}+13.3125NL$$

Regarding the estimated effects (Table 3), the NL (1 L to 5 L) promoted a positive effect on Abs, increasing by 27 g·m^−2^. The C_CH_ also showed a significant positive effect, increasing the Abs to 15 g·m^−2^. The responses of the generated surfaces demonstrate that the increase in the C_CH_ and the NL applied an increase to the Abs values, as can be seen in Figure 2c. The C_AC_ increase does not generate a significative variation in the Abs response of the materials, as shown in Figure 2b,d,f.

Materials formed with five and three layers of coatings, except for COA11 (C_CH_ = 1.0%, C_AP_ = 1.8%, C_AC_ = 0.2, NL = 5) and COA15 (C_CH_ = 1.0%, C_AP_ = 1.8%, C_AC_ = 1.2, NL = 5), showed high Abs, which could be associated with the hydrophilic nature of chitosan. The paper’s hygroscopic matrix reduces the material’s mechanical properties (resistance) in the presence of water [49]. According to Bordenave et al. [50], chitosan coating maintains the hydrophilic characteristic of paper, even at low concentrations, and is responsible for controlling the interactions with water molecules.

According to Table 2, the coated paperboard with NL = 1 L presented the lowest Abs. The highest C_PA_ (1.8% *w*/*w*) promoted a negative and significant effect in the Abs, reducing to 5 g·m^−2^, which is associated with the hydrophobic character of palmitic acid. The cellulose surface modification improves the hydrophobic performance of paper properties for packaging applications, being able to extend the application of cellulosic packaging even in humid conditions or in contact with food products with a high water activity [51,52]. The palmitic acid in the coating formulation can enhance the water resistance of the cellulose network of the paperboard. Similar results were obtained in a beeswax-milk protein coating, indicating that lipids on the coating surface reduced the water absorption capacity of the cellulosic material [53]. It is very well known that the water barrier properties of natural polymers are less effective than synthetic polymers. However, there is an environmental urgency for alternatives to reduce polluting packaging materials.

The performance of coated packaging materials is strictly related to the mass transport properties of the material, which is influenced by the type, composition, and thickness of the coating [54]. By analyzing the effects (Table 3), the level of variation of all the individual variables, from −1 to +1, promoted a negative and significant effect in the WVTR of the coated paperboard. Increasing the C_CH_ (1.0 to 2.0% *w*/*w*) and C_PA_ (0.2 to 1.8% *w*/*w*), the WVTR reduced to 10 and 9 g·d^−1^·m^−2^, respectively. Increasing the coating layers NL = 1 L to 5 L, promoted a significant reduction in WVTR to 49 g·d^−1^·m^−2^ of coated paperboard. The increased heterogeneity of the coatings generated by their constituents may have caused a more tortuous path for the permeation of water vapor molecules, influencing the WVTR response.

Coatings with higher solid content presented the lowest WVTR values. According to Song, Xiao, and Zhao [55], the coating solution fills the porous structure of the paper surface, reducing the cellulosic pores and reflecting directly on the water vapor permeability of the material. Other natural polymers showed the same characteristic, such as sodium caseinate, which according to Khwaldia [56], reduced the water vapor permeability by 75% (CW = 18 g·m^−2^), 69% (CW = 12 g·m^−2^), 60% (CW = 10 g·m^−2^), 32.7% (CW = 6 g·m^−2^), and 25% (CW = 3 g·m^−2^), when compared to uncoated paper.

The higher the concentration of activated carbon promoted a negative effect in the WVTR of coated paperboards, in the order of 15 g·d^−1^·m^−2^. Activated carbon has a non-polar surface that preferentially forms bonds with other non-polar substances, such as hydrophobic compounds [57]. The water in contact with the activated carbon may have been repelled and consequently reduced the WVTR of the materials. Moreover, closing the paper’s pores, the coating applied to the paperboard surface must present a resistance to the polarity of the water vapor, preventing interactions between the polar groups of cellulosic fibers and water [58].

The fat permeability of the coated materials indicated that the high concentration of chitosan (C_CH_ = 2.0% *w*/*w*) increased the number of the Kit solution adopted as the fat repellency value. The COA10 sample (C_CH_ = 2.0%, C_PA_ = 0.2%, C_AC_ = 0.2% *w*/*w* and NL = 5 L) had a Kit repellency value equal to 10, while COA14 (C_CH_ = 2.0%, C_PA_ = 0.2%, C_AC_ = 1.2%, and NL = 5 L) and COA16 (C_CH_ = 2.0%, C_PA_ = 1.8%, C_AC_ = 1.2%, and NL = 5 L) had a total fat barrier (Table 2).

The ability of chitosan to act as a fat barrier is associated with cationic groups in its structure (NH_3_^+^), interacting electrostatically with the anionic groups present in the lipids, retaining the fat and preventing its permeation into the cellulosic matrix [59]. The presence of more layers also contributes to the fat barrier. Kopacic et al. [60] observed that chitosan-based coating applied in two layers on the surface of the paperboard with different drying techniques obtained a high resistance to lipids.

According to the Factorial Experimental Design, a microstructural analysis was performed to evaluate the effects of the coatings on the paper morphology with distinct layers (1 L or 5 L). The SEM was performed on two samples containing the same formulation (C_CH_ = 2.0% *w*/*w*, C_PA_ = 1.8% *w*/*w*, and C_AC_ = 1.2% *w*/*w*), and with different numbers of layers. Figure 2 shows the SEM of COA8 (NL = 1 L) and COA16 (NL = 5 L) on the cross-section of the materials.

The highest number of layers (5 L) fills the paperboard matrix’s pores with the coating deposition in the void spaces between the cellulosic fibers (Figure 3b). According to Tanpichai et al. [61] the chitosan coating first fills the pores of the cellulosic matrix and the formation of the chitosan film on the surface of the paper occurs later, when all the pores are filled. According to Fernandes et al. [62], the increase in the number of chitosan coating layers contributes to filling the pores and covering the paper’s surface due to the film-forming capacity of chitosan. Li and Rabnawaz [63] observed that chitosan coating on the paper surface formed a material with smoother cellulosic fibers, filling the pores on the paper surface. A coating of chitosan incorporated with PDMS (polydimethylsiloxane) generated a more uniform material with no visible pores.

The highest C_CH_ (2.0% *w*/*w*) improved the water barrier property of the coated paperboard (decreased the WVTR significantly) and the fat barrier due to its cationic characteristic and supported the presence of palmitic acid and activated carbon in the coating polymer matrix. The C_AC_ = 1.2% (*w*/*w*) promoted a negative and significant effect on the water vapor barrier (decreasing the WVTR of coated papers) due to its polarity that repelled water molecules. The C_AP_ = 1.8% (*w*/*w*) provides hydrophobic characteristics and reduces the water vapor permeability rate of the coated paperboard. Five layers of coating (NL = 5 L) significantly improved the water barrier (WVTR), fat permeability, and pore filling of the cellulosic matrix of the paperboard due to the greater solid total of the coating. However, the application of 5 L influenced the processability of the material.

The number of layers contributes to the barrier properties (WVTR and fat), filling cellulosic pores. Three chitosan layers (3 L, the central point of the Factorial Experimental Design) would be enough to fill the paperboard surface pores. Then there would be a saturation of the coating in the paper matrix [64].

The Experimental Factorial Design 2^4^ plus central points was used as a tool to evaluate the effect of chitosan, palmitic acid, and activated carbon concentrations (independent variable) on the final properties of the coating paperboard, searching to optimize each independent variable in the final formulation of the coating material. The coating formulation containing the C_CH_ = 2.0% *w*/*w*, C_PA_ = 1.8% *w*/*w*, and C_AC_ = 1.2% *w*/*w* and applied to NL = 3 L on the surface of the paperboard was selected due to its potential application as a packaging material, and its improved water vapor and fat barrier properties based on the estimated effect responses of each independent variable studied.

## 4. Conclusions

The facile, low-cost, non-toxic, and sustainable coated paperboard formulation based on chitosan, palmitic acid, and activated carbon applied in multiple layers on the paperboard surface showed the ability to act as a barrier to water vapor and fat, which is suitable for food-contact applications. The chitosan concentration of 2.0% (*w*/*w*) filled the pores of the cellulosic matrix of the paperboard. The palmitic acid concentration of 1.8% (*w*/*w*), activated carbon of 1.2% (*w*/*w*), and the application of three layers improved the moisture barrier of the coated paperboard. Chitosan coating can act as a fat barrier due to its cationic characteristic. The application of three layers enhanced the properties of the paperboard. The coating formulation indicated the potential for incorporating activated carbon into a biopolymer matrix, enhancing its application in packaging systems that requires a restriction to gas permeation and single-use packaging, for example those used for food delivery and for dry and high-fat foods. Using natural polymers as a coating on paper surfaces for packaging has shown a potential eco-friendly material that can form and be applied in single-used packaging, reducing the environmental impact.

## Figures and Tables

**Figure 1 foods-11-04037-f001:**
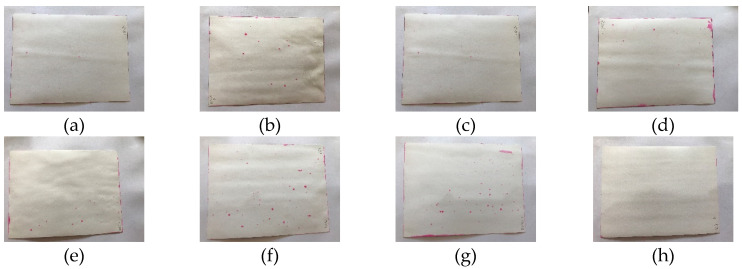
View of the back of the paperboard sheets to assess the homogeneity of the coatings obtained from Factorial Experimental Design 2^4^ + central points. (**a**) COA1. (**b**) COA2. (**c**) COA3. (**d**) COA4. (**e**) COA5. (**f**) COA6. (**g**) COA7. (**h**) COA8. (**i**) COA9. (**j**) COA10. (**k**) COA11. (**l**) COA12. (**m**) COA13. (**n**) COA14. (**o**) COA15. (**p**) COA16. (**q**) COA17. (**r**) COA18. (**s**) COA19.

**Figure 2 foods-11-04037-f002:**
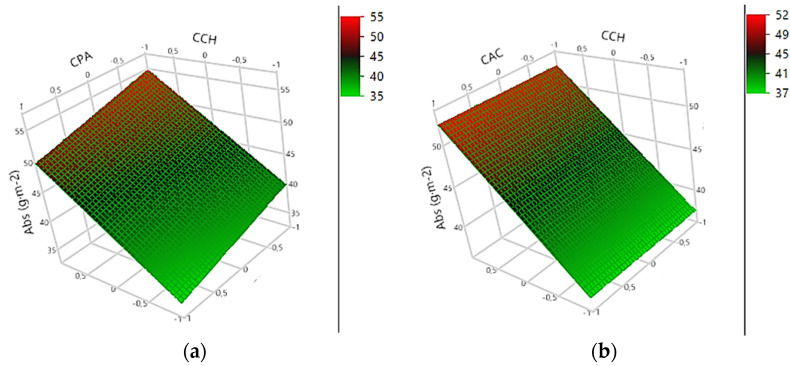
Response surface for water absorption capacity (Abs): (**a**) C_CH_ × C_PA_; (**b**) C_CH_ × C_AC_; (**c**) C_CH_ × NL; (**d**) C_AC_ × C_PA_; (**e**) C_PA_ × NL and (**f**) NL × C_AC_.

**Figure 3 foods-11-04037-f003:**
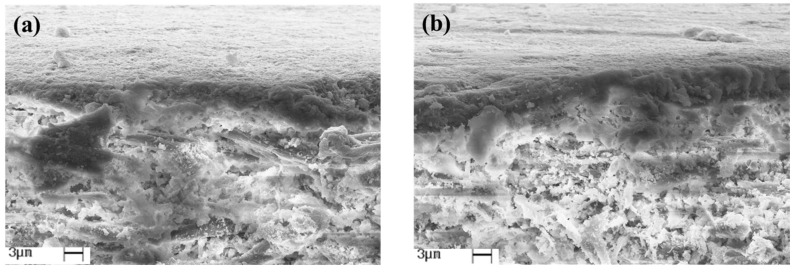
Microstructural analysis of materials by SEM at 3000× magnification in the cross-section of coated paperboard samples: (**a**) COA8 (NL = 1 L) and (**b**) COA16 (NL = 5 L).

**Table 1 foods-11-04037-t001:** Variables and levels of independent variables of coatings of chitosan, palmitic acid, and activated carbon, according to the Factorial Experimental Design 2^4^ + central points.

Variable	Level		
	−1	0	1
C_CH_ (% *w*/*w*)	1.0%	1.5%	2.0%
C_PA_ (% *w*/*w*)	0.2%	1.0%	1.8%
C_AC_ (% *w*/*w*) *	0.2%	0.7%	1.2%
NL	1 L	3 L	5 L

* based on chitosan mass.

**Table 2 foods-11-04037-t002:** Coating weight (CW), thickness, grammage, water absorption capacity (Abs), water vapor transmission rate (WVTR), and fat resistance of coated paperboard with different formulations obtained according to Factorial Experimental Design 2^4^ + central points.

Run	Nomenclature	CW (g·m^−2^)	Independent Variables				Coating Paperboard Properties				
			C_CH_ (*w*/*w*)	C_PA_ (*w*/*w*)	C_AC_ (*w*/*w*)	NL	Thickness (µm)	Grammage (g·m^−2^)	Abs (g·m^−2^)	WVTR (g·d^−1^·m^−2^)	Fat Resistance Kit Number
1	COA1	2.2	1.0% (−1)	0.2% (−1)	0.2% (−1)	1 L (−1)	364 ± 13	262 ± 9	28 ± 1	184 ± 11	7
2	COA2	4.0	2.0% (+1)	0.2% (−1)	0.2% (−1)	1 L (−1)	371 ± 11	264 ± 11	30 ± 1	175 ± 10	7
3	COA3	5.1	1.0% (−1)	1.8% (+1)	0.2% (−1)	1 L (−1)	358 ± 15	258 ± 10	26 ± 1	221 ± 13	3
4	COA4	6.9	2.0% (+1)	1.8% (+1)	0.2% (−1)	1 L (−1)	386 ± 8	271 ± 2	29 ± 1	187 ± 14	7
5	COA5	2.2	1.0% (−1)	0.2% (−1)	1.2% (+1)	1 L (−1)	341 ± 8	255 ± 10	30 ± 1	190 ± 4	4
6	COA6	4.1	2.0% (+1)	0.2% (−1)	1.2% (+1)	1 L (−1)	364 ± 14	258 ± 11	36 ± 2	185 ± 10	8
7	COA7	5.1	1.0% (−1)	1.8% (+1)	1.2% (+1)	1 L (−1)	365 ± 12	250 ± 2	29 ± 1	186 ± 10	3
8	COA8	7.0	2.0% (+1)	1.8% (+1)	1.2% (+1)	1 L (−1)	382 ± 14	263 ± 7	35 ± 2	163 ± 15	5
9	COA9	11.0	1.0% (−1)	0.2% (−1)	0.2% (−1)	5 L (+1)	361 ± 11	260 ± 3	66 ± 4	154 ± 6	6
10	COA10	20.1	2.0% (+1)	0.2% (−1)	0.2% (−1)	5 L (+1)	361 ± 8	261 ± 2	62 ± 2	158 ± 10	10
11	COA11	25.6	1.0% (−1)	1.8% (+1)	0.2% (−1)	5 L (+1)	427 ± 18	267 ± 2	33 ± 3	148 ± 7	3
12	COA12	34.7	2.0% (+1)	1.8% (+1)	0.2% (−1)	5 L (+1)	444 ± 18	279 ± 3	75 ± 4	125 ± 10	8
13	COA13	11.1	1.0% (−1)	0.2% (−1)	1.2% (+1)	5 L (+1)	379 ± 19	265 ± 4	49 ± 3	150 ± 6	4
14	COA14	20.3	2.0% (+1)	0.2% (−1)	1.2% (+1)	5 L (+1)	372 ± 12	266 ± 3	69 ± 3	134 ± 8	Total barrier
15	COA15	25.7	1.0% (−1)	1.8% (+1)	1.2% (+1)	5 L (+1)	428 ± 25	271 ± 2	28 ± 2	101 ± 43	6
16	COA16	34.9	2.0% (+1)	1.8% (+1)	1.2% (+1)	5 L (+1)	448 ± 59	271 ± 2	74 ± 4	126 ± 23	Total barrier
17	COA17	13.7	1.5% (0)	1.0% (0)	0.7% (0)	3 L (0)	393 ± 17	260 ± 2	52 ± 2	142 ± 7	6
18	COA18	13.7	1.5% (0)	1.0% (0)	0.7% (0)	3 L (0)	382 ± 17	260 ± 1	50 ± 3	150 ± 7	5
19	COA19	13.7	1.5% (0)	1.0% (0)	0.7% (0)	3 L (0)	383 ± 38	257 ± 3	54 ± 3	162 ± 2	6

**Table 3 foods-11-04037-t003:** Average of the estimated effects and analysis of variance (ANOVA) for dependent variables (thickness, grammage, water absorption capacity, and water vapor transmission rate properties) of coating paperboard based on Factorial Experimental Design.

	PROPERTIES			
	Thickness (μm)	Grammage (g·m^−2^)	Abs (g·m^−2^)	WVTR (g·d^−1^·m^−2^)
Mean	385 *	264 *	44	162 *
Curvature (Central Point)	2	−10 *	17 *	−20 *
C_CH_	12 *	6 *	15 *	−10 *
C_PA_	42 *	5 *	−5 *	−9 *
C_AC_	0	−3 *	0	−15 *
NL	35 *	8 *	27 *	−49 *
C_CH_ × C_PA_	6 *	4 *	9 *	−4
C_CH_ × C_AC_	1	−1	5 *	6
C_CH_ × NL	−4	−2	11 *	8 *
C_PA_ × C_AC_	0	−2	1	−12 *
CP_A_ × NL	26 *	4 *	−4 *	−15 *
C_CH_ × NL	9 *	4 *	−4 *	−4
C_CH_ × C_PA_ × C_AC_	−1	−2	−3 *	9 *
C_CH_ × C_PA_ × NL	5 *	−1	9 *	7 *
C_CH_ × C_AC_ × NL	−2	−2	3 *	2
C_PA_ × C_AC_ × NL	−6 *	−1	0	7 *
Determination coefficient (R^2^)	0.82	0.58	0.98	0.80
*p*-value	0.13	0.18	1.73 × 10^−8^ *	0.02 *
Lack of fit (F test)	449	63	215	1140
F-value	2.25	1.79	35.00 *	5.41 *
F-tabulated	3.84	3.84	3.84	3.98

*—Statistically significant (*p* < 0.05).

## Data Availability

The data is contained in the article.

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
