# Peer review of "Sustainable Coating Paperboard Packaging Material Based on Chitosan, Palmitic Acid, and Activated Carbon: Water Vapor and Fat Barrier Performance"

_foods, 2022, doi:10.3390/foods11244037_

Round 1

Reviewer 1 Report

Dear Authors,
You present here a type of paper coating that you suggest as an alternative to plastic-based packaging.
First of all, a basic remark: If you consider your test materials with water vapor permeation rates between about 100 and 220 g/(m² day) and compare them with other coated papers, including aluminum laminates, then you must surely notice that orders of magnitude lie in between in terms of functionality. You would thus have to clearly highlight which types of packaging you can and cannot substitute with your experimental materials.
Further comments:
Lines 74 - 77: You can't prevent or really control the permeation of water vapor and normal gases, unlike trace substances like ethylene, with physically absorbent materials, just delay them a bit.
Lines 95 - 98: Please write in complete sentences!
Lines 113 - 119: Please describe the experimental procedures in more detail (concentrations, specific substances) and do not simply refer to another publication. This is not pleasing for a reader.
Lines 125 - 126: Can you say whether the successive coatings may have dissolved the previous ones?
Lines 185 - 187: Explain the meaning of all parameters in the table heading.
Line 205, images: Explain that the pictures show the coated side of the specimens
Table 3 and text from lines 207 to 336: Here they interpret phenomenologically the results of their statistical analyses. What is almost completely missing here are the basic physical dependencies, for example, that the permeation rate behaves reciprocally to the thickness of a barrier layer. These are well-known dependencies that should also be used and presented. I consider an interpretation and discussion based purely on statistical correlations to be unscientific as long as there are known correlations.

Author Response

Dear

Please find attached the revised version of manuscript foods-2080818Sustainable coating paperboard packaging material based on chitosan, palmitic acid, and activated carbon: water vapor and fat barrier performance”, by Jackson Wesley Silva dos Santos, Vitor Augusto dos Santos Garcia, Anna Cecilia Venturini, Rosemary Aparecida de Carvalho, Classius Ferreira da Silva, Cristiana Maria Pedroso Yoshida to Foods.

We would like to thank the valuable comments made by the Editor and we really appreciated the time dedicated to manuscript revision. The article was reviewed carefully in accordance with the guidance document and the comments. All changes and comments were highlighted in manuscript.

Thank you again for the opportunity to disseminate our findings in this prestigious journal.

Sincerely yours,

Dear Authors,
You present here a type of paper coating that you suggest as an alternative to plastic-based packaging.

01) First of all, a basic remark: If you consider your test materials with water vapor permeation rates between about 100 and 220 g/(m² day) and compare them with other coated papers, including aluminum laminates, then you must surely notice that orders of magnitude lie in between in terms of functionality. You would thus have to clearly highlight which types of packaging you can and cannot substitute with your experimental materials. Further comments:

Author’s comment: We agree with the reviewer's comment. The application of natural polymers as packaging materials aims to partially replace materials that are difficult to biodegrade. A paper coated with aluminum has final properties of magnitudes much greater than a paper coated with chitosan and palmitic acid.

However, the biodegradation rate of laminated paper no longer follows that of uncoated paper. It may be that the separation of multilayer materials becoming more expensive and difficult to process in some cases. The urgency in the search for sustainable alternatives has become important by trying a partial substitution in food products that do not require restricted properties. 

It was inserted a potential application of the material developed in the conclusion.

The coating formulation indicated the potential for incorporating activated carbon into a biopolymer matrix, enhancing its application in packaging systems that requires a retard in gas permeation, single-use packaging, for example those used for food delivery and for dry and high-fat foods.02) Lines 74 - 77: You can't prevent or really control the permeation of water vapor and normal gases, unlike trace substances like ethylene, with physically absorbent materials, just delay them a bit.

Author’s comment: It was revised. The “controlled” expression has been replaced in the main text.03) Lines 95 - 98: Please write in complete sentences!

Author’s comment: It was revised.04) Lines 113 - 119: Please describe the experimental procedures in more detail (concentrations, specific substances) and do not simply refer to another publication. This is not pleasing for a reader.

Author’s comment: We understand the importance of making methodologies easier to read and reproduce. A citation has been inserted for the full description of the methodology. The experimental methodology was described after the citation.

The chitosan, palmitic acid, and activated carbon coating dispersion were prepared following Yoshida, Oliveira, and Franco [29]. The chitosan dispersion was prepared at different concentrations in an acidic medium. Acetic acid was added stoichiometrically, considering the chitosan's mass and degree of acetylation. The suspension was homogenized in magnetic stirring for 60 min. According to the Experimental Design, palmitic acid and activated carbon were incorporated at different concentrations under rigorous homogenization at 20,000 rpm (UltraTurrax homogenizer, T25, IKA, Germany) for 10 min.

The independent variables (chitosan, palmitic acid, active carbon concentrations) were used according to the Factorial Experimental Design, and are presented in Table 1 and Table 2, including the number of layers applied.05) Lines 125 - 126: Can you say whether the successive coatings may have dissolved the previous ones?

Author’s comment: This phenomenon was not visually observed in the successive applications of the chitosan, palmitic acid and activated carbon solution. The drying process was done right after the coating solution application.06) Lines 185 - 187: Explain the meaning of all parameters in the table heading.

Author’s comment: The parameters presented in Table 2 has been described in the previous paragraph and in the heading of Table.07) Line 205, images: Explain that the pictures show the coated side of the specimens

Author’s comment: It was revised.08) Table 3 and text from lines 207 to 336: Here they interpret phenomenologically the results of their statistical analyses. What is almost completely missing here are the basic physical dependencies, for example, that the permeation rate behaves reciprocally to the thickness of a barrier layer. These are well-known dependencies that should also be used and presented. I consider an interpretation and discussion based purely on statistical correlations to be unscientific as long as there are known correlations.

Author’s comment: The statistical analysis was carried out following the Factorial Experimental Design tool. The main statistical tool application was the evaluation of coating formulations based on the independent variables’ effects (chitosan, palmitic acid, activated carbon concentrations and number of layers) promoted in the final properties of the developed material. 

The effects were observed such as the cationic characteristic of chitosan that acts in the barrier to fat, the higher solids content could fill the cellulosic matrix of the paper, the hydrophobic and non-polar characteristic increased a moisture barrier.

Reviewer 2 Report

This study aimed to develop and characterize a sustainable alternative material based on paperboard packaging coated with chitosan, palmitic acid, and activated carbon, applied in several layers, and evaluate its performance in the water vapor and fat barrier.

Firstly, the paper is well organized; by the way, I recommend the authors just to improve the quality of Figure 2, and if possible, please provide the SEM analysis for the surface analysis. Additionally, my suggestion to authors is to add the updated literature in 2022

What is the optimum point of the model, and what is the actual,  Is it fit the actual to the optimum model ?

What is the meaning of the (- )value for Table 3? What could be the reason for the high lack of fit, and low R2 for the grammage response?

For some references, doi is not provided.

Author Response

Dear

Please find attached the revised version of manuscript foods-2080818Sustainable coating paperboard packaging material based on chitosan, palmitic acid, and activated carbon: water vapor and fat barrier performance”, by Jackson Wesley Silva dos Santos, Vitor Augusto dos Santos Garcia, Anna Cecilia Venturini, Rosemary Aparecida de Carvalho, Classius Ferreira da Silva, Cristiana Maria Pedroso Yoshida to Foods.

We would like to thank the valuable comments made by the Editor and we really appreciated the time dedicated to manuscript revision. The article was reviewed carefully in accordance with the guidance document and the comments. All changes and comments were highlighted in manuscript.

Thank you again for the opportunity to disseminate our findings in this prestigious journal.

Sincerely yours,

This study aimed to develop and characterize a sustainable alternative material based on paperboard packaging coated with chitosan, palmitic acid, and activated carbon, applied in several layers, and evaluate its performance in the water vapor and fat barrier.

01) Firstly, the paper is well organized; by the way, I recommend the authors just to improve the quality of Figure 2, and if possible, please provide the SEM analysis for the surface analysis. Additionally, my suggestion to authors is to add the updated literature in 2022

Author’s comment: Figure 2 was improved. Unfortunately, the superficial SEM images were not present visual differences. It was not possible to verify the differences in the microstructure. The literature was updated.

02) What is the optimum point of the model, and what is the actual, Is it fit the actual to the optimum model?

Author’s comment: The water absorption capacity model was significant and predictive. The Experimental Design was applied to verify the effects of the independent variables on the final properties of coated card paper, defining the best formulation depending on the application of coated card paper. The optimal point was not the main object of this work.

03) What is the meaning of the (-) value for Table 3? What could be the reason for the high lack of fit, and low R2 for the grammage response?

Author’s comment: The negative values (-) presented in Table 3 indicated that the variation in the level from -1 to +1 of a certain independent variable caused a decrease in the response of the evaluated properties, which may have been significant (indicated by the *) or not. For example, increasing the activated carbon concentration from 0.2% to 1.2% significantly decreases the water vapor transmission rate of the material in 15 g·d-1·m-2, contributing to improve the water vapor barrier. The description of the Table 3 was revised in the manuscript, explaining the estimated mean effects. The variation in statistical responses for grammage may be associated with the nature of natural polymers. The intermolecular interactions of biopolymers strongly influence the responses of their properties. The difference in the number of layers applied, varying from one to five layers, may also have influenced the weight of the coating applied to the surface of the paperboard. The discussion was revised in the manuscript.

04) For some references, doi is not provided.

Author’s comment: It was revised. The available doi in the databases were inserted.

Reviewer 3 Report

Dear Authors,

Please find in the attachment my comments and suggestion.

Kind Regards.

Author Response

Dear

Please find attached the revised version of manuscript foods-2080818Sustainable coating paperboard packaging material based on chitosan, palmitic acid, and activated carbon: water vapor and fat barrier performance”, by Jackson Wesley Silva dos Santos, Vitor Augusto dos Santos Garcia, Anna Cecilia Venturini, Rosemary Aparecida de Carvalho, Classius Ferreira da Silva, Cristiana Maria Pedroso Yoshida to Foods.

We would like to thank the valuable comments made by the Editor and we really appreciated the time dedicated to manuscript revision. The article was reviewed carefully in accordance with the guidance document and the comments. All changes and comments were highlighted in manuscript.

Thank you again for the opportunity to disseminate our findings in this prestigious journal.

Sincerely yours,

In this study “Sustainable coating paperboard packaging material based on chi-tosan, palmitic acid, and activated carbon: water vapor and fat barrier perfor-mance”, the authors investigated the use of palmitic acid and activated carbon applied on paperboard surfaces as a sustainable alternative to improve the moisture and fat barrier properties of cellulosic packaging material. The topic is interesting but the first part of the manuscript is difficult to follow due to a lack of appropriate comments on tables 2 and 3 and figure 1. In the following I report other comments and suggestions:

01) In line 85, consider adding a comment to explain clearly the novelty of the work with respect to other works in the literature.

Author’s comment:  It was inserted.

02) Consider the line from 176 to 178, to support the sentence “The range of concentrations of independent variables was defined based on preliminary experimental analysis” with more information also in the supporting information.

Author’s comment: It was revised. The determination of the values adopted in the independent variables studied was determined according to preliminary tests carried out in the study itself. The preliminary tests were developed based on the available results found in the literature.

03) In Tables 2 and 3, and Figure 1 please consider providing comments to support or resume the table meaning.

Author’s comment: It was revised.
